# End-to-End Autonomous Navigation Based on Deep Reinforcement Learning with a Survival Penalty Function

**DOI:** 10.3390/s23208651

**Published:** 2023-10-23

**Authors:** Shyr-Long Jeng, Chienhsun Chiang

**Affiliations:** 1Department of Mechanical Engineering, Lunghwa University of Science and Technology, Taoyuan City 333326, Taiwan; 2Department of Mechanical Engineering, National Yang Ming Chiao Tung University, Hsinchu City 300093, Taiwan; chienhsunchiang.en10@nycu.edu.tw

**Keywords:** actor–critic (AC) method, autonomous, reinforcement learning (RL), wheeled mobile robots (WMRs)

## Abstract

An end-to-end approach to autonomous navigation that is based on deep reinforcement learning (DRL) with a survival penalty function is proposed in this paper. Two actor–critic (AC) frameworks, namely, deep deterministic policy gradient (DDPG) and twin-delayed DDPG (TD3), are employed to enable a nonholonomic wheeled mobile robot (WMR) to perform navigation in dynamic environments containing obstacles and for which no maps are available. A comprehensive reward based on the survival penalty function is introduced; this approach effectively solves the sparse reward problem and enables the WMR to move toward its target. Consecutive episodes are connected to increase the cumulative penalty for scenarios involving obstacles; this method prevents training failure and enables the WMR to plan a collision-free path. Simulations are conducted for four scenarios—movement in an obstacle-free space, in a parking lot, at an intersection without and with a central obstacle, and in a multiple obstacle space—to demonstrate the efficiency and operational safety of our method. For the same navigation environment, compared with the DDPG algorithm, the TD3 algorithm exhibits faster numerical convergence and higher stability in the training phase, as well as a higher task execution success rate in the evaluation phase.

## 1. Introduction

Autonomous intelligent mobile robots [1] are suitable for completing various tasks in dangerous and complex environments; such tasks include urban rescue, public security patrol, and epidemic prevention. Motion planning, in which information obtained from the external environment by sensors is used to evaluate the optimal collision-free path between a starting point and an ending point, is an essential technique in autonomous navigation. Traditional motion planning [2] is a core component of the pipeline framework and must be integrated with other subtasks—such as perception, localization, and control—to accomplish autonomous driving tasks; these tasks [3] are often inflexible and require substantial computational resources and numerous manual heuristic functions because they involve real-time sensory information. The pipeline framework for autonomous driving consists of many interconnected modules, and in the end-to-end method, the entire framework is treated as a single learning task. In end-to-end autonomous motion [2,4], raw sensor data are directly used as the input to a neural network, which outputs low-level control commands. This method is attractive because complex modules do not need to be designed; instead, a network with a simple structure is constructed, and the entire motion process is optimized into a single machine-learning task.

On the basis of their principles and era of development, robot planning methods can be divided into traditional and learning-based methods. Traditional methods include graph search algorithms, sample-based algorithms, interpolating curve algorithms, and reaction-based algorithms [1]. The Dijkstra method [5] is based on a greedy and optimal graph search; however, it lacks directionality during pathfinding. In the A* method [6], a heuristic function is employed to measure the distance between the real-time search location and the target location; this process results in a targeted search and a higher search speed than that achieved in the Dijkstra method. A rapidly exploring random tree (RRT) [7] is a sample-based approach that generates a sequence of dynamically feasible kinematic connections. The RRT method is sensitive to the sampling distribution, and no guarantee exists that the time taken to converge to the optimal solution would be sufficiently short. The interpolating curve algorithm draws a smooth path based on computer-aided geometric design. Typical path smoothing and curve generation rules include line and circle [8], clothoid curves [9], polynomial curves [10], Bezier curves [11], and spline curves [12]. The potential field method (PFM), the velocity obstacle method (VOM), and the dynamic window approach (DWA) are three reaction-based algorithms that are widely used in engineering and manufacturing. The PFM [13] uses the gradient of a potential field to move the robot from an initial to a target point. The VOM [14] relies on current positions and velocities of robots and obstacles to calculate a reachable avoidance velocity (RAV) space. A proper avoidance velocity is selected from the RAV to avoid static and moving obstacles. The DWA [15] is about selecting the appropriate translational and rotational velocity to maximize an objective function within a dynamic window, which is an online collision avoidance strategy. The shortcoming of traditional methods is that they require knowledge of the environment prior to planning. These methods are unsuitable for solving path-planning problems in environments with unknown characteristics. In addition, when a traditional method is used, trajectory optimization must be performed at the back end after the path search has been performed at the front end, and trajectory optimization requires considerable calculations.

Reinforcement learning (RL) [16,17,18] is a goal-directed computational approach. In contrast to supervised and unsupervised learning methods, RL involves using the reward of interacting with an unknown dynamic environment as a feedback signal, rather than using many labeled samples [19]. RL methods can be categorized into value-based, policy-based, and actor–critic (AC)-based frameworks. Value-based RL is the simplest RL method, and it performs well in most discrete action spaces. In policy-based RL, stochastic policies are optimized by directly mapping states to actions through a probability function. The AC-based framework [20] combines the advantages of value-based and policy-based frameworks and typically includes actor and critic networks. The actor is a policy network that maps input states to output actions, and the critic is a value network that evaluates the quality for each state–action pair.

An RL agent learns by directly interacting with the environment without supervision and without a model of the environment. At the end of an episode, the agent generally receives positive rewards through environmental feedback. For long-distance navigation in environments containing obstacles, robots have difficulty obtaining the final positive reward signal. The sparse reward problem [21] leads to slow learning and difficult convergence. Reward shaping [22] is a means of manually tuning and modifying fine-grained reward signal values for robots in different states. The tuning that is performed in reward shaping is intuitive and highly dependent on the expert experience of the person conducting the process. Inappropriate rewards might lead to changes in the optimal policy and produce abnormal behavior. In the curiosity-driven method [23], the sparse reward problem is solved by exploiting existing trajectories. An intrinsic curiosity module extracts additional intrinsic reward signals from the environment and encourages the agent to explore effectively. Hindsight experience replay (HER) [24] is another approach used for addressing the sparse reward problem. In HER, additional reward signals are explored and acquired during training by mapping the no-reward state to new targets and by replacing previous targets. Another approach for addressing reward sparsity is hierarchical RL (HRL) [25,26], in which the original task is hierarchically decomposed into multiple continuous and easy-to-solve subtasks, which are then further divided and completed to provide dense reward signals to the agent. The reward function is difficult to configure for some specific and complex planning tasks. In inverse RL (IRL) [27], expert trajectories are utilized for inversely learning the reward function, and policies are then optimized. IRL is an imitation-learning paradigm [28], as is behavior cloning, which relies on a supervised learning process and suffers from a mismatch between the participant’s strategy and the expert’s strategy.

AC-based deep RL (DRL) approaches—including the deep deterministic policy gradient (DDPG) [29], twin-delayed DDPG (TD3) [30], and soft AC (SAC) [31] algorithms—are often used to optimize action sequences for robots. Vecerik et al. [32] proposed a general and model-free DDPG-based approach for solving high-dimensional robotics problems. Other studies [33,34] have proposed DDPG-based path-planning methods in which HER is used to overcome the performance degradation caused by sparse rewards. TD3 with traditional global path planning was employed in [35] to improve the generalization of the developed model. In [30] and [36], the TD3 and HRL methods were combined, the state–action space was divided in accordance with the information maximization criterion, and a deterministic option policy was then learned for each region of the state–action space. The SAC algorithm [31,37] uses a stochastic policy and obtains the optimal policy by optimizing the entropy. HER is used in the SAC algorithm to improve sampling efficiency and results in favorable exploration performance.

In an environment exposed to multiple obstacles, the agent learns how to avoid obstacles and reach its destination without collision. The DWA [15] is a traditional method of static obstacle avoidance. Due to complex conditions and equations, dynamic obstacles require a large amount of computing time to predict the next movement of the obstacle. CADRL [38] and MRCA [39] are representative obstacle avoidance methods based on reinforcement learning. CADRL aims to avoid pedestrians and needs to obtain pedestrian position, speed, and body shape information. The disadvantage is that it cannot be avoided if the pedestrian is not detected. MRCA can avoid various dynamic obstacles through the distance information of LiDAR without detecting dynamic objects. In a multi-agent learning environment, it is relatively difficult for MRCA to comprehensively avoid dynamic obstacles through multiple different avoidance strategies or no avoidance strategies. Liang et al. [40] used multiple perception sensors including 2-D LiDAR and depth cameras for smooth collision avoidance. Choi et al. [41] proposed a framework in decentralized collision avoidance in which each agent independently makes its decision without communication with others. A collision avoidance/mitigation system [42] was proposed to rapidly evaluate potential risks associated with all surrounding vehicles and to maneuver the vehicle into a safer region when faced with critically dangerous situations through a predictive occupancy map. A two-stage RL-based collision avoidance approach [43] was proposed. The first stage is a supervised training method with a loss function that encourages the agent to follow the well-known reciprocal collision avoidance strategy. The second stage is using a traditional RL training method to maximize reward function and refine the policy. A hybrid algorithm [44] of RL and force-based motion planning was presented to solve the distributed motion-planning problem in dense and dynamic environments. A new reward function provides conservative behavior by changing collision punishment to decrease the chance of collision in crowded environments. Many studies [26,40,41,45] adopt the method of imposing a large penalty on the reward when the robot collides with an obstacle or approaches an obstacle. The survival penalty imposes a negative reward at each time step, which encourages the robot to reach the goal as quickly as possible. Obstacle avoidance based solely on reinforcement learning may lead to path-finding problems in some cases. To solve this problem and improve navigation efficiency, the path planner is integrated with reinforcement learning-based multi-obstacle avoidance.

To address the severe problem of reward sparsity in complex environments, survival penalties can be applied as rewards to ensure successful learning. A survival penalty-based approach is used within an AC framework to solve a motion-planning problem in which the target position and orientation are aligned simultaneously. We design a comprehensive reward function and conduct it in three navigation scenarios to ensure that the goal can be approached while avoiding obstacles dynamically.

The remainder of this paper is organized as follows. Section 2 details the navigation problem and the requirements for a WMR. Section 3 briefly describes two AC frameworks, namely, DDPG and TD3. In the aforementioned section, we elaborate on the observation state, action space, and reward function and then construct the environment and agent for DRL. Section 4 presents details on the proposed model’s training and evaluation and on simulation results for four scenarios. Section 5 discusses the rationality of the survival penalty function and the potential applications of this study. Finally, Section 6 provides the conclusions of this study.

## 2. Problem Statement and Preliminaries

A path is a route from one point in space to another. The path-planning problem is closely related to the collision avoidance problem. Although most studies have aimed to solve the path-planning problem in a flexible and efficient manner, they have considered only motion planning involving position transitions, without considering orientation-related tasks. In this paper, we propose a planning approach in which orientation and localization tasks are tackled simultaneously for further investigating the potential of WMRs. We address attitude adjustment in the final goal of the WMR’s task. As illustrated in Figure 1, our goal is for the WMR to move in a timesaving manner and without colliding with obstacles while possessing the required orientation. The proposed method can be applied to many situations. For example, the operation of an automated parking system is a position and orientation colocalization problem. The system must locate the target parking space, define a path that avoids obstacles, and park in the space with the correct orientation and high accuracy [46]. The proposed method involves the planning of collision-free paths during precise directional navigation. To meet requirements for position and orientation transformation, kinematic constraints as well as position and attitude constraints are considered.

As depicted in Figure 1, a differential-drive WMR consists of two rear driving wheels parallel to the back-to-front axis of the body and a caster wheel that supports the robotic platform. The WMR is moved by two direct motors driving the wheels. The global inertia is fixed in the plane of motion, and the moving WMR has a local body frame. The global inertial frame and local body frame are denoted by X–Y and x–y, respectively. The origin of the local frame’s coordinate system is the center of gravity of the WMR. The x-direction is the direction that the WMR is facing, whereas the y-direction points to the left if the WMR is facing forward. The WMR is assumed to be in contact with a nondeformable horizontal plane on which it purely rolls and never slips during motion.

The attitude of the WMR is described by the following generalized coordinate vector with five components: $q={\left[X_{c},\;Y_{c},\;\psi ,\theta_{R},\;\theta_{L}\right]}^{T}\in R^{5}$, where $\left(X_{c},\;Y_{c}\right)$ are the inertial coordinates of the WMR’s center of mass, ψ is the yaw angle of the WMR relative to the horizontal inertial axis X, and $\theta_{R}$ and $\theta_{L}$ are the rotation angles of the right and left driving wheels, respectively. The width of the WMR is 2b, the distance between the center of mass and geometric center of the WMR platform is d, and the driving wheel radius is r. As shown in Figure 1, the centroid $P_{C}$ of the WMR does not coincide with its geometric center $P_{O}$. The WMR is subject to three nonholonomic constraints when its wheels are rolling and not slipping.
***A***(***q***)***q*** = 0,(1)
where
$$A\left(q\right)=\left[\begin{array}{ccccc} -\sin\psi & \cos\psi & -d & 0 & 0\\ \cos\psi & \sin\psi & b & -r & 0\\ \cos\psi & \sin\psi & -b & 0 & -r \end{array}\right]$$
is the nonholonomic constraint matrix.

Let ${\dot{\theta }}_{R}$ and $\dot{{\;\theta }_{L}}$ represent the angular velocities of the right and left driving wheels, respectively. If $\omega \left(t\right)={\left[{\dot{\theta }}_{R},\;\dot{{\;\theta }_{L}}\right]}^{T}$ is considered a control input, the kinematic model of the WMR [47] can be expressed as follows:(2)$$\dot{q}=S\left(q\right)\omega \left(t\right)$$
where
$$S\left(q\right)=\frac{1}{2}\left[\begin{array}{cc} r\cos\psi -\frac{d\cdot r}{b}\sin\psi & r\cos\psi -\frac{d\cdot r}{b}\sin\psi \\ r\sin\psi -\frac{d\cdot r}{b}\cos\psi & \mathrm{rsin}\psi -\frac{d\cdot r}{b}\cos\psi \\ \frac{r}{b} & \frac{-r}{b}\\ 2 & 0\\ 0 & 2 \end{array}\right].$$

The line velocity and angular velocity of the WMR at point $P_{C}$ are denoted by v and ω, respectively. The relationship between v, ω, ${\dot{\theta }}_{R}$, and $\dot{{\;\theta }_{L}}$ is as follows:(3)$$\left[\begin{array}{c} {\dot{\theta }}_{R}\\ \dot{{\;\theta }_{L}} \end{array}\right]=\left[\begin{array}{cc} \frac{1}{r} & \frac{b}{r}\\ \frac{1}{r} & \frac{-b}{r} \end{array}\right]\left[\begin{array}{c} v\\ \omega \end{array}\right].$$

Equation (3) is substituted into (2) to obtain the ordinary form of a WMR with two actuated wheels, which is expressed as follows:(4)$$\dot{q}=J\left(q\right)\left[\begin{array}{c} v\\ \omega \end{array}\right],$$
where
$$J\left(q\right)=\left[\begin{array}{cc} \cos\psi & -d\cdot \sin\psi \\ \sin\psi & d\cdot \cos\psi \\ 0 & 1\\ \frac{1}{r} & \frac{b}{r}\\ \frac{1}{r} & \frac{-b}{r} \end{array}\right],$$
which satisfies the equation A(q)J(q)=0.

## 3. Reinforcement Learning

### 3.1. DDPG Algorithm

As shown in Figure 2, the DDPG algorithm [29] combines a replay buffer, a deterministic policy, and an AC architecture to solve decision problems in a continuous task space. The term $\theta^{Q}$ represents the parameter of the critic network, which approximates the state–action value $Q(s,\;a|\theta^{Q})$, whereas $\theta^{\mu }$ is the parameter of the actor network, which approximates the deterministic policy $\mu (s|\theta^{\mu })$. The actor outputs the probability distribution of an action, whereas the critic uses the state–action value $Q(s,\;a|\theta^{Q})$ to evaluate the performance of the actor and guide the actor’s next action.

The objective function in the DDPG algorithm is maximization of the expected cumulative reward $R\left(\tau \right)$ in episode τ; this function is expressed as follows:(5)$$J\left(\theta^{Q}\right)=E_{\tau \sim \pi_{\theta }(\tau )}[R(\tau )]\approx E[Q(s,\;a|\theta^{Q})]$$

In the DDPG approach, Monte Carlo-based updating is converted into a temporal difference (TD) method. The actual reward is compared with the critic’s estimate to obtain the TD error, which is used to adjust the parameters of the critic network and thus obtain a more accurate estimate. When Monte Carlo methods are used, the actual reward cannot be determined until the end of the RL episode. In the TD method, only the next interaction state $s_{t+1}$ must be reached before the TD target can be formed and $Q\left(s_{t},a_{t}|\theta^{Q}\right)$ can be updated. The TD target value $y_{t}$ at time step t is expressed as follows:(6)$$y_{t}=r_{t}+\gamma Q^{target}\left(s_{t+1},\;\mu^{target}\left(s_{t+1}|\theta^{\mu^{target}}\right)|\theta^{Q^{target}}\right),$$
where $r_{t}$ represents the current reward; $s_{t+1}$ represents the state in time step *t* + 1; $\theta^{Q^{target}}$ and $\theta^{\mu^{target}}$ are copies of $\theta^{Q}$ and $\theta^{\mu }$, respectively, and the target networks for low-frequency updates; $Q^{target}(\cdot )$ is the state–action value obtained by the critic target network; and $\mu^{target}(\cdot )$ is the action obtained by the actor target network. The actor selects an action by using the estimate of $Q[s,\;\mu \left(s|\theta^{\mu }\right)|\theta^{Q}]$ obtained by the critic network; thus, the estimate should be as close as possible to the target value.

For the TD method, the objective function of the critic network that minimizes the mean square error (MSE) between the target value $y_{t}$ and the estimated value $Q(s_{t},\;\mu \left(s_{t}|\theta^{\mu }\right)|\theta^{Q})$ can be expressed as follows:(7)$$J(\theta^{Q})=E\left[{\left(y_{t}-Q\left(s_{t},\;\mu \left(s_{t}|\theta^{\mu }\right)|\theta^{Q}\right)\right)}^{2}\right]$$

The parameter $\theta^{Q}$ of the critic network is optimized using the gradient descent algorithm. In this case, the gradient is calculated as follows:(8)$$\nabla_{\theta^{Q}}J\left(\theta^{Q}\right)\approx E\left[\left(y_{t}-Q\left(s_{t},\;\mu \left(s_{t}|\theta^{\mu }\right)|\theta^{Q}\right)\right)\cdot \nabla_{\theta^{Q}}Q\left(s_{t},\;\mu \left(s_{t}|\theta^{\mu }\right)|\theta^{Q}\right)\right]$$

Conversely, the parameter $\theta^{\mu }$ of the actor network is optimized using the gradient ascent algorithm. In this case, the gradient is computed as follows:(9)$$\nabla_{\theta^{\mu }}J\left(\theta^{\mu }\right)\approx E\left[\nabla_{\theta^{\mu }}Q\left(s_{t},a|\theta^{\mu }\right)\right]=E\left[\nabla_{\theta^{\mu }}\mu (s_{t},\theta^{\mu })\cdot \nabla_{\theta^{Q}}Q\left(s_{t},\;\mu \left(s_{t}|\theta^{\mu }\right)|\theta^{Q}\right)\right]$$

The gradient is computed using the chain rule as the expected value of the product between the gradient of the critic network with respect to the state–action and the gradient of the actor network with respect to its action. The actor is updated to make the output action be more favorably evaluated by the critic.

In the DDPG algorithm, a replay buffer is used to break the Markov nature of sampled data, and two target networks are employed to ensure the stability of the training process. The TD method is used to redefine the objective functions of the critic network and actor network as follows:(10a)$$J\left(\theta^{Q}\right)=\frac{1}{N}\sum_{t}{\left(y_{t}-Q\left(s_{t},\;\mu \left(s_{t}|\theta^{\mu }\right)|\theta^{Q}\right)\;\right)}^{2},$$
(10b)$$J\left(\theta^{\mu }\right)=\frac{1}{N}\sum_{t}Q\left(s_{t},a|\theta^{\mu }\right),$$
where N is the number of $<s_{t},\;a_{t},\;r_{t},s_{t+1}>$ tuples sampled from the replay buffer. The objective gradients for the critic and actor networks are expressed as follows:(11a)$$\nabla_{\theta^{Q}}J\left(\theta^{Q}\right)=\nabla_{\theta^{Q}}\frac{1}{N}\sum_{t}{\left(y_{t}-Q\left(s_{t},\;\mu \left(s_{t}|\theta^{\mu }\right)|\theta^{Q}\right)\right)}^{2}$$
(11b)$$\nabla_{\theta^{\mu }}J(\theta^{\mu })≅\frac{1}{N}\sum_{i}\nabla_{\theta^{\mu }}\mu \left(s_{i},\;\theta^{\mu }\right)\cdot \nabla_{\theta^{Q}}Q\left(s_{t},\;\mu \left(s_{t}|\theta^{\mu }\right)|\theta^{Q}\right)$$

The parameters $\theta^{Q}$ and $\theta^{\mu }$ of the critic and actor networks, respectively, are updated as follows:(12a)$$\theta_{i+1}^{Q}=\theta_{i}^{Q}-\alpha_{Q}\nabla_{\theta^{Q}}J\left(\theta^{Q}\right),$$
(12b)$$\theta_{i+1}^{\mu }=\theta_{i}^{\mu }+\alpha_{\mu }\nabla_{\theta^{\mu }}J(\theta^{\mu }),$$
where $\alpha_{Q}$ and $\alpha_{\mu }$ are the learning rates of the critic and actor networks, respectively. The target functions of the critic and actor networks are updated at low frequency toward the main networks as follows:(13a)$$\theta_{i+1}^{Q^{target}}=\tau \theta^{Q}+(1-\tau )\theta_{i}^{Q^{target}},$$
(13b)$$\theta_{i+1}^{\mu^{target}}=\tau \theta^{\mu }+(1-\tau )\theta_{i}^{\mu^{target}},$$
where τ is a smoothing factor. These new target functions are obtained using a soft update method that improves the stability of network convergence.

### 3.2. TD3 Algorithm

A common mode of failure for the DDPG algorithm involves the learned *Q* function beginning to overestimate the *Q* values substantially. Errors of the *Q* function are introduced into the training of the actor network, which leads to policy violation. The TD3 algorithm [30], which is an improvement of the DDPG algorithm, can reduce overestimation of the value function. In the TD3 algorithm, delayed actor updates, dual critics and actors, and additional clip noise are employed to control actions. As shown in Figure 3, two critic functions, namely, $Q_{1}\left(s,a|\theta^{Q_{1}}\right)$ and $Q_{2}\left(s,a|\theta^{Q_{2}}\right)$, are learned concurrently through MSE minimization in almost the same way that the single critic function is learned in the DDPG algorithm. The predicted value of $y_{t}$ that is selected is the smaller of the predictions of the two target critics:(14)$$y_{t}=r_{t}+\gamma \;{min}_{j=1,2}Q_{j}^{target}\left(s_{i+1},\;clip\left(\mu^{target}\left(s_{i+1}|\theta^{\mu^{target}}\right)\right)|\theta^{Q_{j}^{target}}\right),$$
where *j* represents the two target critics, and the clip function constrains future actions within the lower and upper bounds specified for the controls. As an output, the clip function returns its argument unless this argument violates the bound, in which case the argument is set to be equal to the bound. Each critic computes the MSE of the TD error e  within the batch as follows:(15)$$MSE\;e^{2}=\frac{1}{N}\sum_{t}{\left(y_{t}-{min}_{j=1,2}Q_{j}(s_{t},a_{t}|\theta^{Q_{j}})\right)}^{2},$$
where N is the batch sampled from the replay buffer R. The lowest prediction from a pair of critics is employed to prevent overestimation of the value function. The parameters of the critic functions $\theta^{Q_{1}}$ and $\theta^{Q_{2}}$ are updated through one step of gradient descent as follows:(16)$$\nabla_{\theta^{Q_{j}}}J\left(\theta^{Q}\right)=\nabla_{\theta^{Q}}\frac{1}{N}\sum_{t}{\left(y_{t}-{min}_{j=1,2}Q_{j}\left(s_{t},\;a_{t}|\;\theta^{Q_{j}}\right)\right)}^{2}.$$

Using a small critical value of the target and regressing to that value helps to prevent overestimation of the Q function. The parameters of the target actor and target critic are updated less frequently in the TD3 algorithm than in the DDPG algorithm.

### 3.3. Setting of the State and Action

The motion planning required to perform a given WMR task can be fundamentally divided into navigation and obstacle avoidance. The navigation module obtains information on the relative attitude of the WMR with respect to the target, whereas the anti-collision planning module receives raw LiDAR sensor data that contain information on the unknown and dynamic environment in which the WMR is moving [48]. The WMR’s system uses the aforementioned information for agent training and to determine the WMR’s next action. The current pose and target pose are denoted as $\left(X_{c},Y_{c},\;\psi \right)$ and $\left(X_{c,goal},Y_{c,goal},\;\psi_{goal}\right)$, respectively (Figure 4a). To speed up the training and improve the convergence stability of the algorithm, the pose state vector $s_{pos}$ is set to be a six-dimensional vector normalized in the range [−1, 1]. This vector is expressed as follows:
(17)$$s_{pos}={\left[\frac{x_{rel}}{d_{rel,max}},\;\frac{y_{rel}}{d_{rel,max}},\frac{d_{rel}}{d_{rel,max}},\frac{\psi_{2,rel}}{\pi },\frac{\psi_{3,rel}}{\pi },\frac{\psi_{4,rel}}{\pi }\right]}^{T},$$
where $x_{rel}$ and $y_{rel}$ are the coordinates of the WMR’s current position in the body coordinate system based on the WMR’s target position, $d_{rel}$ is the distance between the current position and the target position, $d_{rel,max}$ is the maximum $d_{rel}$ value, $\psi_{2,rel}$ is the azimuth angle of viewing the current WMR from the tail of the target, $\psi_{3,rel}$ is the angle of viewing the target from the first-person perspective in WMR, and $\psi_{4,rel}$ is the relative yaw angle with respect to the target. These parameters are expressed as follows:(18a)$$d_{rel}=\sqrt{{\left(X_{c,goal}-X_{c}\right)}^{2}+{\left(Y_{c,goal}-Y_{c}\right)}^{2}}$$
(18b)$$\psi_{1,rel}={tan2}^{-1}\left(\frac{Y_{c,goal}-Y_{c}}{X_{c,goal}-X_{c}}\right)$$
(18c)$$\psi_{2,rel}=\psi_{1,rel}-\psi_{goal}-\pi$$
(18d)$$x_{rel}=d_{rel}\cdot \cos(\psi_{2,rel}+\pi )$$
(18e)$$y_{rel}=d_{rel}\cdot \sin(\psi_{2,rel}+\pi )$$
(18f)$$\psi_{3,rel}=\psi_{1,rel}-\psi -\pi$$
(18g)$$\psi_{4,rel}=\psi -\psi_{goal}$$

As the WMR successfully navigates itself to the target, all elements in the pose state vector $s_{pos}$ approach 0.

To plan an obstacle-free path, the robot’s system must extract information from the environment by using sensor data and thereby generate commands that ensure obstacle avoidance. Figure 4b shows an example of LiDAR data obtained in the horizontal plane. Laser scanning is performed between 120° and −120°, and sensors are established to detect reflectance of laser beams at intervals of 12°. If the sensor does not detect any object within a certain distance, the distance of the nearest obstacle is considered to be equal to the maximum detectable distance $d_{max,LiDAR}$; otherwise, the length is the distance between the WMR and the detected obstacle. The range information of LiDAR is normalized into the range [0, 1] and is expressed as follows:(19)$$s_{LiDAR}={\left[\frac{d_{0,\;detect}}{d_{max,LiDAR}},\frac{d_{1,\;detect}}{d_{max,LiDAR}},\ldots ,\frac{d_{20,\;detect}}{d_{max,LiDAR}}\;\right]}^{T}$$
where $d_{i,LiDAR}$ is the length detected by the *i*th laser sensor. The input state information $s_{t}$ at time step t includes the pose information $s_{pos}$ relative to the target and the obstacle information $s_{LiDAR}$ relative to the obstacle; this information comprises the following set:(20)$$s_{t}={\left[s_{pos},s_{LiDAR}\right]}^{T}$$

The action to be taken by the WMR is the set of the linear forward velocity v and the angular velocity ω around the centroid of the WMR.
(21)$$a_{t}={\left[v,\omega \right]}^{T}$$

### 3.4. Survival Penalty Function

When the reward function is used as a training metric to encourage or discourage action, this function has a strong influence on learning performance. In many studies [19,26,49], the distance $d_{rel}$ between the current position and the target position has been employed as the main parameter of the reward function. When a positive reward design is used, the closer the WMR is to the target, the higher the reward. Once the agent successfully reaches the target area, it obtains the highest reward. In the illustration presented in Figure 5a, the WMR is initially at the left side and reaches the final position on the right side. The maximum distance $d_{max}$ for a straight track is set as 10 intervals. The positive reward $r_{pos}$ is proportional to the distance from the starting point and is expressed as follows:(22)$$r_{pos}=d_{max}-d_{rel}.$$

Path planning is performed at regular intervals. When the WMR is traveling at a forward velocity of 1 and 2 intervals per time step, the agent receives a cumulative reward of 55 and 30, respectively, after completing episodic work. The goal in DRL is to guide an agent that is attempting to maximize the cumulative reward from the initial state distribution. A potential drawback of only implementing positive rewards is that the agent employs a slow movement policy to perform the situational task and thereby obtains a relatively high cumulative reward. In study [18], auxiliary negative rewards have also been assigned to penalize near-standstill action when the WMR is traveling below a certain speed.

The survival penalty function in which a negative relative distance $d_{rel}$ is used as the reward for straight-line forward motion is defined as follows:(23)$$r_{neg}=-d_{rel}$$

As displayed in Figure 5b, the highest penalty is awarded at the starting point, which is the farthest point from the target, whereas the lowest penalty is awarded at the end point. For forward velocities of 1 and 2 intervals per time step, the cumulative rewards are −55 and −30, respectively. The agent essentially moves the WMR from the starting point to the destination and thus completes the linear motion task as quickly as possible. The linear motion condition can be implemented by adopting the maximum driving speed strategy. During turning maneuvers, a smaller turning radius can be produced by reducing the driving speed and altering the rotation angle. Survival penalty functions have the intrinsic advantage of guiding the agent to complete the motion task in the fewest number of steps; thus, the decisions made are different to those made in positive reward design.

To guide the WMR precisely into the target area and simultaneously achieve the correct position and orientation, the intersecting $x_{rel}$–$y_{rel}$ space of the coordinate plane is roughly divided into four quadrants, as illustrated in Figure 6. In human driving behavior, a vehicle is driven into the correct position by approaching the target from the rear. We first consider the case in which the WMR is behind the target, that is, $x_{rel}<0$. We define a locking angle $\psi_{lock}$ as follows:(24)$$\psi_{lock}=\max\left(\left\lfloor \psi_{2,rel}\right\rfloor ,\left\lfloor \psi_{3,rel}\right\rfloor ,\left\lfloor \psi_{4,rel}\right\rfloor \right),$$
where $\psi_{2,rel}$, $\psi_{3,rel}$, and $\psi_{4,rel}$ are calculated using (18c), (18f), and (18g), respectively. When the locking angle $\psi_{lock}$ is less than 36°, the agent might force the robot to head toward the locking angle and then accurately approach the desired orientation and position (Figure 6a). The smaller the locking angle, the smaller the penalty awarded. When the locking angle is greater than 36° but less than 90°, the main goal of the agent is to keep the WMR away from the $y_{ref}$-axis by enforcing a clearage distance $\left|x_{lock1}\right|$ and to move the WMR progressively closer to the $x_{ref}$-axis. When the locking angle is greater than 90°, $x_{rel}$ remains in the range [$x_{lock2}$, $x_{lock1}$], and a change-of-orientation motion is performed to turn the vehicle toward the destination. To prevent the clearage distance from being too small or too large for large orientation adjustments, the two boundary distances $x_{lock1}$ and $x_{lock2}$ are set as −60 and −90, respectively. The negative reward when the WMR is in the second or third quadrant is as follows:(25)$$r_{penalty1}=\left\{\begin{array}{l} -\frac{d_{rel}}{d_{rel,max}}-\frac{\left\lfloor y_{rel}\right\rfloor }{d_{rel,max}}-\frac{\psi_{lock}}{\pi }\;\;\;\;\;\;\;\;\;\;\mathrm{if}\;\psi_{lock}<18{}^{\circ}\;\;\;\;\;\;\;\;\;\;\;\;\;\;\;\;\;\\ \;\;\;\;\;-0.5-\frac{\left\lfloor y_{rel}\right\rfloor }{d_{rel,max}}-\frac{\psi_{lock}}{\pi }\;\;\;\;\;\;\;\;\;\;\;{if\;\psi }_{lock}<36{}^{\circ}\;\;\;\;\;\;\;\;\;\;\;\;\;\\ {\;\;\;\;\;\;\;\;\;\;\;\;\;\;\;\;\;\;\;\;\;\;\;\;\;\;\;\;\;\;\;\;\;\;\;\;\;\;\;\;\;\;\;\;\;\;\;\;\;\;\;\;\;\;if\;\;\psi }_{lock}\leq 90{}^{\circ}\;\\ -0.8-\frac{\left\lfloor y_{rel}\right\rfloor }{d_{rel,max}}-\frac{x_{rel}-x_{lock1}}{-x_{lock1}}\;\;\;\;\;\;\;\;\;\;\;\;\;\;\;\;\;\;\;\mathrm{if}\;x_{rel}\geq x_{lock1}\;\;\;\;\;\;\;\\ \;\;\;\;\;-0.8-\frac{\left\lfloor y_{rel}\right\rfloor }{d_{rel,max}}\;\;\;\;\;\;\;\;\;\;\;\;\;\;\;\;\;\;\;\;\;\;\;\;\;\;\;\;\;\;\;\;\mathrm{other}\;\;\;\;\;\;\;\;\;\;\;\;\;\;\;\;\;\;\;\\ {\;\;\;\;\;\;\;\;\;\;\;\;\;\;\;\;\;\;\;\;\;\;\;\;\;\;\;\;\;\;\;\;\;\;\;\;\;\;\;\;\;\;\;\;\;\;\;\;\;\;\;\;\;\;\;if\;\;\psi }_{lock}>90{}^{\circ}\;\\ \;\;\;-1-\frac{\psi_{lock}}{\pi }-\frac{x_{rel}-x_{lock1}}{-x_{lock1}}\;\;\;\;\;\;\;\;\;\;\;\;\;\;\;\;\;\;\;\;\;\;\;\;\mathrm{if}\;x_{rel}\geq x_{lock1}\\ \;\;-1-\frac{\psi_{lock}}{\pi }-\frac{x_{rel}-x_{lock2}}{x_{lock1}}\;\;\;\;\;\;\;\;\;\;\;\;\;\;\;\;\;\;\;\;\;\mathrm{if}\;x_{rel}\leq x_{lock2}\;\;\;\;\;\\ \;\;\;\;\;-1-\frac{\psi_{lock}}{\pi }\;\;\;\;\;\;\;\;\;\;\;\;\;\;\;\;\;\;\;\;\;\;\;\;\;\;\;\;\;\;\;\;\;\;\;\;\;\;\mathrm{other}\;\;\;\;\;\;\;\;\;\;\;\;\;\;\;\; \end{array}\right.$$

The constant negative bias in the first term of each penalty equation results in the WMR being primarily steered in the correct direction, with the lock angle being reduced and the punishment lowered as the WMR approaches the target.

The second case we consider is that in which the WMR is located in the first or fourth quadrant. The objective of the reward function in this case is to guide the WMR away from the dangerous passageway adjacent to the target area and into the rear of the target. When the alignment angle $\left|\psi_{4,rel}\right|$ is greater than 144°, the front of the WMR is pointed away from the positive $x_{ref}$-axis direction and slightly toward the rear of the target. The size of the main penalty is determined by the values of $\left|\psi_{4,rel}\right|$ and $x_{rel}$, and this penalty guides the WMR to move to the second or third quadrant. An auxiliary penalty is added to prevent the WMR from passing through the dangerous passageway close to the $x_{ref}$-axis. The clear distance of $y_{way}$ is set as 60. When the WMR is facing the positive $x_{ref}$-axis—that is, $\left|\psi_{4,rel}\right|\leq \;144{}^{\circ}$—the WMR performs a U-turn in the first or the fourth quadrant, which causes $\left|\psi_{4,rel}\right|$ to increase. The negative reward when the WMR is in the first or the fourth quadrant is expressed as follows:(26)$$r_{penalty2}=\left\{\begin{array}{l} \;\;\;\;\;\;\;\;\;\;\;\;\;\;\;\;\;\;\;\;\;\;\;\;\;\;\;\;\;\;\;\;\;\;\;\;\;\;\;\;\;\;\;\;\;\;\;\;\;\;\;\;\;\;\;\;\;\;\mathrm{if}\;\left|\psi_{4,rel}\right|>144{}^{\circ}\;\\ -2+\frac{\left|\psi_{4,rel}\right|}{\pi }-\frac{x_{rel}}{d_{rel,max}}-\frac{y_{way}-\left|y_{rel}\right|}{y_{way}}\;\;\;\;\;\;\;\;\;\;\;\;\mathrm{if}\;\left|y_{rel}\right|\leq y_{way}\\ \;\;\;-2+\frac{\left|\psi_{4,rel}\right|}{\pi }-\frac{x_{rel}}{d_{rel,max}}\;\;\;\;\;\;\;\;\;\;\;\;\;\;\;\;\;\;\;\;\;\;\;\;\mathrm{other}\;\;\;\;\;\;\;\;\;\;\;\;\;\;\;\;\;\\ \;\;\;\;\;\;\;\;\;\;\;\;\;\;\;\;\;\;\;\;\;\;\;\;\;\;\;\;\;\;\;\;\;\;\;\;\;\;\;\;\;\;\;\;\;\;\;\;\;\;\;\;\;\;\;\;\mathrm{other}\\ -3+\frac{\left|\psi_{4,rel}\right|}{\pi }-\frac{y_{way}-\left|y_{rel}\right|}{y_{way}}\;\;\;\;\;\;\;\;\;\;\;\;\;\;\;\;\;\;\;\;\;\;\;\;\;\;\;\;\;\;\mathrm{if}\;\left|y_{rel}\right|\leq y_{way}\\ \;\;\;-3+\frac{\left|\psi_{4,rel}\right|}{\pi }\;\;\;\;\;\;\;\;\;\;\;\;\;\;\;\;\;\;\;\;\;\;\;\;\;\;\;\;\;\;\;\;\;\;\;\;\;\;\;\;\;\;\;\mathrm{other}\;\;\;\;\;\;\;\;\;\;\;\;\;\;\;\;\;\; \end{array}\right.$$

The survival penalty functions expressed in (25) and (26) indicate that negative rewards (punishments) are always awarded as the WMR transitions from a high-penalty state to a low-penalty state. An additional reward is provided at the end of each episode: 5 and −10 for successful completion of the navigation task and for a failed mission, respectively.

### 3.5. Collision Avoidance Constraints

During the training phase, when navigation paths are being explored, scenarios involving collisions between the WMR and the environment or obstacles are easily generated. In many studies [26,49], scholars have penalized collision situations by terminating the exploration episode and introducing additional negative rewards. The closer the detected obstacle is, the greater the punishment. By instead applying positive rewards, the agent can avoid unrewarding collision paths as much as possible and continue to explore paths that correctly achieve the navigation goal while accumulating large rewards. However, this strategy is not possible when applying the survival penalty method. One reason for this restriction is that a sudden large penalty might cause the numerical calculation to become unstable and might make it difficult for convergence to be achieved. Another reason is that the agent can easily conclude that a path involving a collision but with few generation steps and a small cumulative penalty is the optimal path, thereby resulting in poor action generation and training failure.

We introduce a simple method that involves exploiting the survival penalty function to manage obstacle avoidance. The TD3 and DDPG algorithms are off-policy algorithms that create experience replay buffers to store historical experiences, randomly sample transitions from these experiences, and then employ these sample data to update the actor and critic networks. The existence of the experience replay buffer helps the agent to learn from previous experiences and improve the efficiency of sample utilization. Figure 7 displays two historical experiences: two consecutive episodes i and i + 1, where episode i involves a collision. The episodes i and  i + 1 consist of n and m steps, respectively, each of which stores an experience $\left[s_{t},\;a_{t},\;r_{t},\;s_{t+1}\right]$ tuple. To increase the cumulative penalty for the collision episode, we can concatenate these two episodes into a new episode. At the end step n of collision episode i, the original next state $s_{end}$ is linked to the state of the first step of episode i + 1. The new episode thus consists of n + m steps. The TD method can be used to increase the cumulative penalty of the original collision episode i  by the cumulative penalty of episode i + 1, whereas the cumulative penalty of episode i + 1 is maintained the same.

## 4. Simulation and Results

### 4.1. Network Parameter Settings

Table 1 provides details on the architectures of the actor and critic networks used in the simulations performed in this study. The six-dimensional relative target information and 21-dimensional laser range findings of these networks are merged into a 27-dimensional input vector. The input layer of the actor network is followed by three fully connected layers, each of which contains 512 nodes and uses the rectified linear unit (ReLU) activation function. The outputs of the output layer include the linear and rotational velocities generated by the hyperbolic tangent function. The range covered by the linear velocity and angular velocity is [−1, 1]. The output actions are multiplied by two scale parameters to determine the final linear and angular velocities that should be directly executed by the WMR. The critic network consists of an input layer, three fully connected layers, and an output layer. Each fully connected layer contains 512 nodes and uses the ReLU activation function. The state–action pair predicts the *Q* value. The input layer of the critic network is the same as that of the actor network. The output of the critic network’s first fully connected layer is concatenated with the output of the actor network. The critic network, the inputs to which are state–action pairs, finally generates a continuous value through a linear activation function.

Table 2 lists the hyperparameter settings employed during the simulation. The discount factor *γ* is set as 0.99, the learning rate of the actor and critic networks is set as 0.00001, and τ = 0.01 for the soft updating of the target networks. The size of the replay buffer is 40,000, and the batch size of updating is 128. Moreover, the maximum number of steps in each episode is set as 300. Simulations are performed on a computer with an Intel i7-11 CPU with 32 GB of memory and an Nvidia RTX 4090 GPU. Python 3.10 is used as the project interpreter. The deep-learning framework PyTorch 2.0 is employed for training the networks in a Windows system.

### 4.2. Virtual Environments

To prove the effectiveness and superiority of the proposed survival penalty function, simulations are performed that involve the DDPG and TD3 algorithms with the same reward functions and hyperparameters. Navigation tasks in four virtual environments are investigated. Table 3 presents the physical parameters of and constraints on the WMR in the learning environment.

#### 4.2.1. Free Obstacle Scenario

In this scenario, we construct a virtual navigation environment in which the initial and target yaw angles have the same magnitude but opposite signs. The distance between the initial and target points is set as 300. The navigation task is accomplished using a survival penalty function. Figure 8 shows the cumulative reward curves when the yaw angle is set as 90° and the DDPG and TD3 algorithms are employed. The red and blue curves generated by the DDPG and TD3 algorithms, respectively, reveal large fluctuations in the estimated cumulative rewards at the beginning of training. The TD3 algorithm converges faster than does the DDPG algorithm; the reward curves of the DDPG and TD3 algorithms become consistent after approximately 400 and 600 episodes of training, respectively. In the stable region (i.e., after the aforementioned numbers of episodes), the mean and variance of the blue curve are slightly lower and smaller, respectively, than those of the red curve. The estimated cumulative reward is considerably larger than the ground-truth cumulative reward, which indicates that the AC framework always overestimates the cumulative reward. Because the TD3 algorithm employs two critic networks, this algorithm exhibits a smaller difference between the ground-truth reward and the estimated reward than does the DDPG algorithm.

Table 4 lists the simulation results obtained for autonomous navigation in virtual environments for various yaw angle pairs, including the estimated cumulative reward, ground-truth cumulative reward, Δ reward, steps per episode, and success rate. The Δ reward column is calculated by subtracting the ground-truth cumulative reward from the estimated cumulative reward. The estimated cumulative reward is always greater than the ground-truth cumulative reward. The autonomous path created by the TD3 algorithm has fewer execution steps than that created by the DDPG algorithm and thus leads to the navigation task being completed more quickly in the same operating environment. Similarly, the navigation success rate achieved using the TD3 algorithm is considerably higher than that achieved using the DDPG algorithm.

Figure 9 displays the autonomous paths planned by the system for initial yaw angles of 0°, 40°, 80°, 120°, 140°, 160°, and 180°. The red and blue curves represent the autonomous paths created by the DDPG and TD3 algorithms, respectively. The green lines radiating from the WMR are the LiDAR scan lines. The paths generated by the TD3 algorithm have small turning radii and involve considerable straight-line movement. The shapes of these paths approximate a section of a rounded rectangle.

Figure 10 shows the trajectories of the normalized forward linear velocity v and angular velocity ω at a yaw angle of 80°. A maximum normalized linear speed of 1 represents maximum forward speed, and a minimum normalized linear speed of −1 represents zero forward speed. Positive and negative normalized angular velocities reveal the directional angular rates of the WMR to the left and right, respectively. The thick black dashed line in Figure 10 shows the evaluation results using the TD3 algorithm when the greedy noise ε is set to zero. The WMR starts at maximum forward linear speed and performs a right turn maneuver. After approximately 16 steps, the WMR adjusts to linear motion, and the normalized angular velocity ω approaches zero. After 36 steps, the WMR gradually turns right again and enters the target area. In the final stages of the navigation task, the WMR reduces linear velocity and varies angular velocity at high frequency to align with the target yaw angle. The WMR navigation path shown in Figure 9b uses 0.5 epsilon greedy noise to increase the path exploration capability. As shown in Figure 10, the fluctuation of the blue line produced by the TD3 algorithm with epsilon greedy noise 0.5 is significantly smaller than that of the red line generated by the DDPG algorithm. The total number of steps for the navigation simulation without/with epsilon greedy noise using the TD3 algorithm is 64 and 69, respectively. The TD3 algorithm has better noise immunity than the DDPG algorithm. In this study, the input state information $s_{t}$ and reward function $r_{t}$ do not refer to the speed information of the WMR. The trajectories of forward linear velocity v and angular velocity ω are not very smooth. Liang [40] imposed a penalized reward associated with angular velocity exceeding a threshold to reduce oscillatory behavior. Chai [26] proposed a hierarchical control framework, which consists of an upper motion-planning layer and a lower tracking layer to form a collision-free algorithm to achieve autonomous exploration with high motion performance. Navigation tasks are suitable for low-speed movements.

#### 4.2.2. Parking Scenario

To demonstrate the general ability of the proposed method to adapt to other environments, we consider a parking scenario. The virtual environment that is simulated consists of two parallel rows of parking spaces in a space with upper and lower boundaries (Figure 11), and partitions exist on both sides of each parking space. The lower-left parking space is considered the starting point, and the remaining seven parking spaces are the target positions. At the origin and destination, the front of the WMR is pointing inside and outside the parking lot, respectively. Figure 11a,b display the autonomous paths planned by the DDPG and TD3 algorithms, respectively, for movement from the fixed initial configuration to the seven parking spaces. In none of the paths does the WMR collide with the partitions on either side of the parking spaces. The WMR can move in a straight line, straight offset, or U-turn and successfully enter the desired parking space.

Table 5 summarizes the statistics for each of the seven autonomous parking scenarios, with 100 evaluations performed using an epsilon greedy noise of 0.5. The DDPG and TD3 algorithms produce almost the same ground-truth cumulative reward for each case. For all episodes, the TD3 algorithm exhibits considerably lower overestimation of cumulative rewards than the DDPG algorithm does. The TD3 algorithm can accomplish path planning with few steps in each navigation round and exhibits a high success rate.

#### 4.2.3. Intersection Scenario

A virtual environment is designed that simulates a general intersection, as illustrated in Figure 12. The WMR starts from entrance 4 of the intersection (on the left in Figure 12), and the four possible destinations are the four exits, namely, the exits that can be reached by turning left, going straight, turning right, or making a U-turn. Figure 12a displays a general two-way intersection, whereas Figure 12b illustrates the same intersection but with an impassable circle in its center. The WMR should not collide with the lane separators or the central part of the roundabout. The red and blue curves displayed in Figure 12 indicate the autonomous paths generated by the DDPG and TD3 algorithms, respectively. In the case of turning left or going straight, the generated paths are considerably different for the two considered types of intersection (i.e., with and without an obstacle). When the agent encounters an obstacle, it passes parallel to the boundary of the obstacle and maintains a certain distance from it. For the intersection containing no obstacle, the autonomous paths generated by the DDPG and TD3 algorithms are almost coincident (Figure 12a), whereas in the environment containing an obstacle, the paths are slightly different (Figure 12b).

Table 6 presents the numerical statistics for the DDPG and TD3 algorithms in the intersection scenario. These algorithms achieve almost the same ground-truth cumulative rewards in each case. For the environment containing an obstacle, the cumulative reward computed using the DDPG algorithm exhibits a large overestimation compared with the ground-truth cumulative reward, which leads to many steps being required to accomplish the navigation task and a low success rate per episode. The TD3 algorithm, which uses the survival penalty function, not only manages complex and different environments but also exhibits higher stability and efficiency than the DDPG algorithm does.

#### 4.2.4. Multi-Obstacle Scenario

The virtual environment is designed to demonstrate the adaptability of our approach to scenarios with multiple obstacles. The scenario shown in Figure 13 has three obstacles. WMR starts from the left side of the figure with a starting yaw angle of 0°. The destination is on the right, but the four possible target yaw angles are set to 0°, 90°, 180°, and 270°. For the first navigation situation where the target yaw angle is 0°, the paths generated by the DDPG and TD3 algorithms pass below and above obstacle 1, respectively, and then enter the target location from the corridor between obstacle 2 and obstacle 3. When the target yaw angle is 90° or 270°, the paths established by the two algorithms almost overlap. The WMR passes through the upper or lower side of obstacles 2 and 3, respectively, and enters the target area. When the target yaw angle is 180°, the red line path generated by DDPG and the blue line path generated by TD3 enter the tail of the target area from above obstacle 2 and below obstacle 3, respectively, perform a U-turn, and successfully complete the navigation task. Although the target position in the navigation situation is the same, different collision-free autonomous paths will be generated sequentially for different target yaw angles. Table 7 summarizes the simulation results of four navigation situations in multi-obstacle scenarios. Due to the vertical symmetry of the multi-obstacle design, the average step size of the 270° target yaw angle calculated by the TD3 algorithm is almost the same as the average step size of the 90° target yaw angle. The TD3 algorithm shows higher stability than the DDPG algorithm.

In order to further verify the actual performance of our method, we conducted comparative simulations using these two algorithms in four virtual environments. A common failure mode of the DDPG algorithm [30] is that the learned Q-function starts to dramatically overestimate Q-values, which then leads to policy failure because it exploits errors in the Q-function. The TD3 algorithm is designed to solve the overestimation bias of the value function by learning two Q functions instead of one Q function. The TD3 algorithm chooses the smaller of the two Q-functions to form the target error loss function. Using a smaller Q value as a target and regressing back to that value helps avoid overestimation in the Q function. To obtain good policy exploration, we add epsilon noise to its action. As shown in Table 4, Table 5, Table 6 and Table 7, the average estimated cumulative reward (Q value) calculated by these two algorithms is always greater than the average ground-truth cumulative reward after 100 episode evaluations. The average estimated cumulative reward calculated by the TD3 algorithm is smaller than the one calculated by the DDPG algorithm. This means that using the smaller target values of the two Q functions of the TD3 algorithm can effectively reduce the impact of estimation errors, improve the average success rate of path planning, and reduce the average number of navigation steps.

In our simulations, the DDPG agent easily forgets previously learned knowledge. During the training process of the obstacle-free scenes, the agent sequentially learns the optimal paths from the configurations of yaw angles of 0°, 20°, …, 180°. When the overall configurations are performed, the autonomous path of the 0° yaw angle configuration becomes poor. The 0° yaw angle configuration needs to be retrained for improving the path quality. Backpropagation, which performs a backward pass to adjust neural network model parameters, may refresh or update previously learned weights. As shown in Figure 9a, the autonomous path derived by the DDPG algorithm still fluctuates slightly, whereas it is a straight line when only training the 0° yaw angle configuration. The memory-forgetting phenomenon causes the success rate of yaw angle configurations of 20° and 180° to be obviously low for DDPG, as shown in Table 4. For multiple obstacle environment configurations, as shown in Figure 13, compared with traditional path-planning methods such as A* and RRT, the agent needs to spend a lot of training time to explore and gradually regress an obstacle-free path. However, the traditional path-planning method is not suitable for yaw angle configurations of 90°, 180°, and 270°.

## 5. Discussion

The reward function is an incentive mechanism that tells the agent what is correct and what is wrong through rewards and penalties. RL applications can be transformed into sequential decision-making problems, and states, actions, and rewards can be established. The agent’s goal is to maximize the total reward. Designing and implementing a reward function that is consistent with an application is challenging. For example, the application of RL in electric vehicle battery-charging management [50] uses positive rewards to allow management to play a major role in coordinating the charging and discharging mechanism and effectively achieve a safe, efficient, and reliable power system. In autonomous navigation applications, as shown in Figure 5, negative rewards can encourage agents to move to the destination as quickly as possible. On the contrary, positive rewards may lead the agent to increase the navigation time (steps) and obtain the maximum reward. In our study, since the navigation time of each navigation situation varies greatly, it is not directly designed as a penalty parameter.

TD3 and DDPG are model-free RL off-policy RL algorithms. Model-free algorithms learn directly from experience and can operate without complete knowledge of environmental dynamics or transition models. Simulated environments are the most common method of training agents. The kinematic model of WMR is based on a two-dimensional framework in the x- and y-directions. When a vehicle travels on uneven terrain, the vehicle will generate significant vertical vibrations. Movements in the z-direction may affect the precise positioning of the vehicle, which is a key requirement for autonomous navigation. Bikes, tripod cars, and four-wheel drive cars are real vehicles with different motion structures. The simulation environment could use an accurate vehicle model to generate relevant experience, but this would require a lot of computer time to solve the nonlinear vehicle equations. The agent relies on sampling to estimate the value function, resulting in noisy estimates and slower convergence, but model-free algorithms can be used in large, complex environments. Our approach can be extended to complex 3D environment applications such as UAVs or robotic manipulators. The goal of path-planning techniques is not only to discover optimal and collision-free paths, but also to minimize various issues such as path length, travel time, and energy consumption that require further research. Using multiple robots to conduct collaborative multi-observation of multiple moving targets is a future research direction.

The main contributions of this study are summarized as follows:(1)Appropriate reward functions are designed by considering human driving experiences and using a survival penalty method so that a wheeled mobile robot (WMR) can receive negative reward feedback at every step. The reward function varies continually on the basis of observation and action. Dense reward signals are found to improve convergence during training, and this finding verifies the feasibility of solving the sparse reward problem encountered in autonomous driving tasks for WMRs.(2)Two consecutive RL episodes are connected to increase the cumulative penalty. This process is performed when the WMR collides with an obstacle. The agent is prevented from selecting the wrong path, learns a path in which obstacles are avoided, and successfully and safely reaches the target.(3)The effectiveness and robustness of the proposed method is evaluated for three navigation scenarios. The simulation results indicate that the proposed method can effectively solve the problem of planning a path along which a WMR can drive to its destination in a complex environment. The results also indicate that the proposed method is robust and self-adaptive.(4)By using the DDPG and TD3 algorithms with the same AC framework, the WMR can safely and quickly reach its target under the simultaneous alignment of position and orientation. For the same environment and standard, compared with the DDPG algorithm, the TD3 algorithm requires considerably fewer computations and has higher real-time performance.(5)The path-planning method has the ability to handle orientation and positioning simultaneously. In an environment without map information, the intermediate waypoints between the departure point and the destination do not need to be pre-determined segmentally, and an end-to-end obstacle-free autonomous U-turn path can be realized. End-to-end vehicle navigation has the potential to be used in high-end industrial robot applications such as electric vehicle parking and forklift picking and placing.

## 6. Conclusions

For the autonomous navigation of a WMR, the target location as well as a target orientation are considered task constraints. Traditional random-action exploration leads to high-probability sparse reward systems. After each action in our method, the agent acquires information regarding the relative position and orientation of the target state, and this information is suitable for providing a dense reward signal, with the provision of a certain immediate reward being a considerably better option than the provision of no reward at all. The survival penalty function utilizes negative rewards to guide the agent to achieve the goal as quickly as possible. The concept underlying the design of the survival penalty function is similar to that underlying human driving behavior, and the negative reward provides a useful means of exploration for the training of agents. Training paradigms based on different task goals can enable the state update policy to be iteratively improved on the basis of the dense feedback signal provided by the environment during the interaction process. Every action taken by the WMR should be rewarded so that it can most effectively assess the quality of its actions and alleviate training difficulty.

The DDPG and TD3 algorithms are deterministic policy gradient algorithms based on the AC framework. In the proposed method, these two algorithms are incorporated with high-precision LiDAR range findings. A process in which two consecutive RL episodes in the experience pool are connected; this process helps the WMR to avoid collisions and enables paths to be planned for environments containing obstacles. The optimal policy can be regarded as a mapping from state to action, which enables the WMR to perform the optimally rewarded actions given the current state of the environment. We conduct simulations for three scenarios to verify the performance of the proposed approach: scenarios involving obstacle-free spaces, a parking lot, and intersections without and with obstacles. The actor network relies heavily on the critic network; thus, the performance of the DDPG algorithm is highly sensitive to critic learning. The DDPG algorithm is unstable because of its sensitivity to its hyperparameters and tends to converge to very poor solutions or even diverge. The TD3 algorithm, which has an auxiliary critic network, more accurately estimates cumulative rewards than does the DDPG algorithm and prevents high levels of cumulative reward overestimation. The simulations conducted in the three scenarios reveal that compared with the DDPG algorithm, the TD3 algorithm converges considerably faster, has a superior convergence effect, and exhibits higher efficiency and adaptability in complex dynamic environments. The TD3 algorithm results in the effective avoidance of obstacles in environments and a high rate of successful task completion.

## Figures and Tables

**Figure 1 sensors-23-08651-f001:**
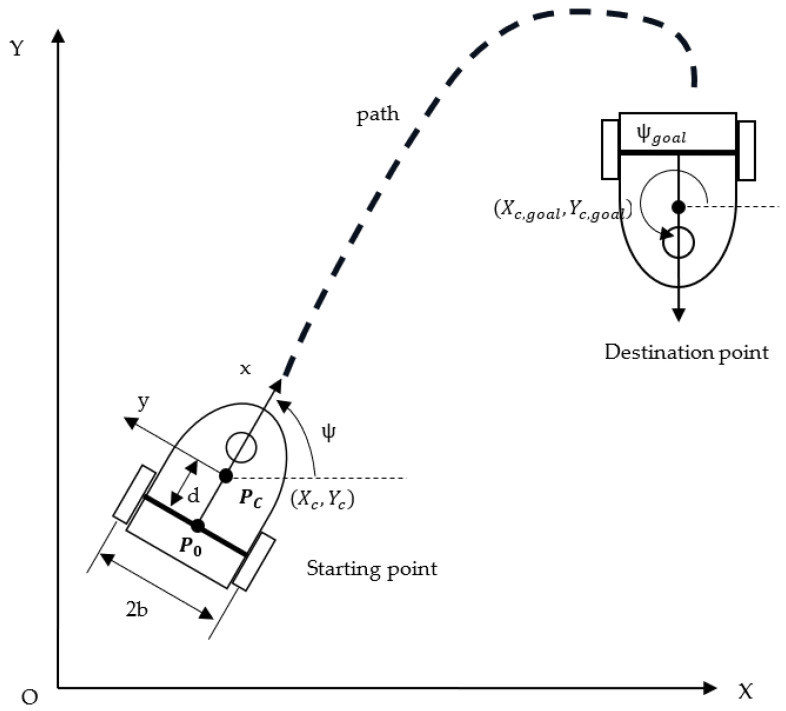
Kinematic model of a WMR.

**Figure 2 sensors-23-08651-f002:**
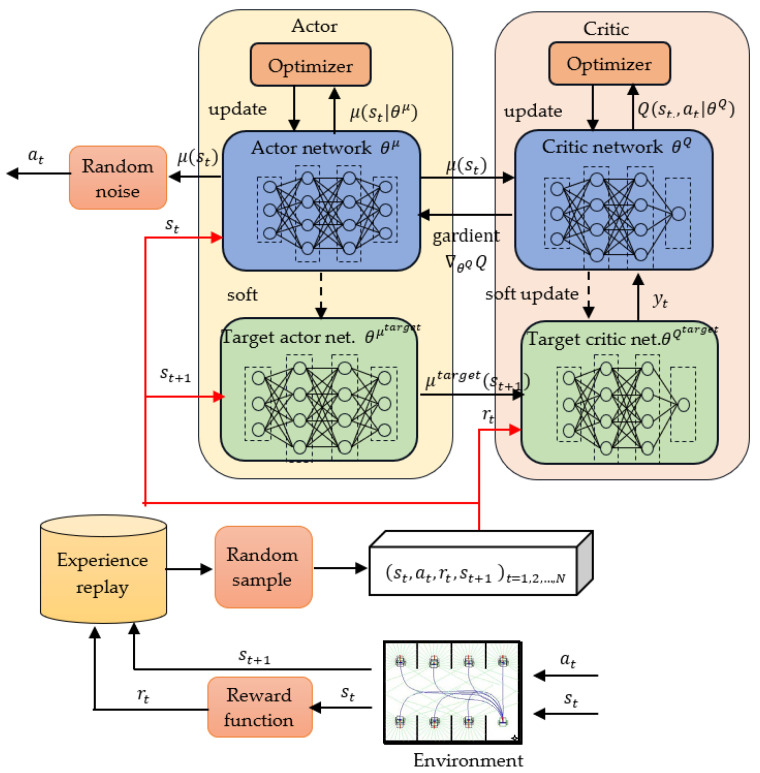
Flowchart of the DDPG algorithm.

**Figure 3 sensors-23-08651-f003:**
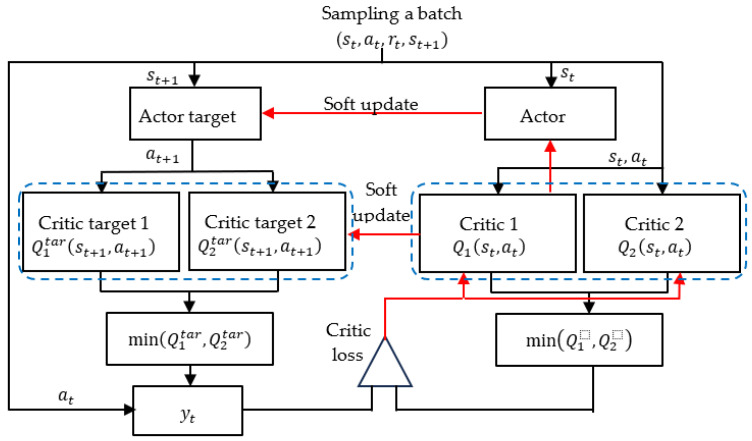
Flowchart of the TD3 algorithm.

**Figure 4 sensors-23-08651-f004:**
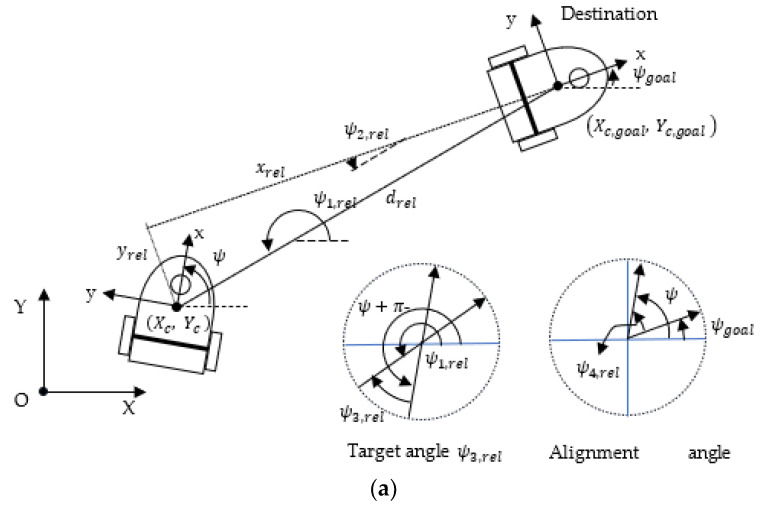
State space of the WMR: (**a**) WMR pose information, (**b**) environmental information provided by a LiDAR sensor scan.

**Figure 5 sensors-23-08651-f005:**
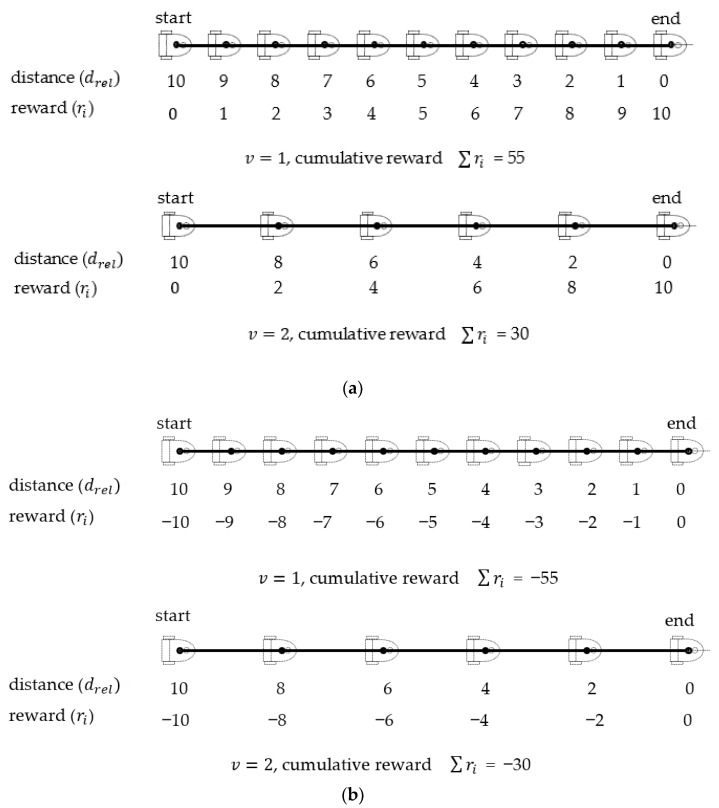
Cumulative rewards of the linear motion using positive and negative rewards: (**a**) positive reward, (**b**) negative reward.

**Figure 6 sensors-23-08651-f006:**
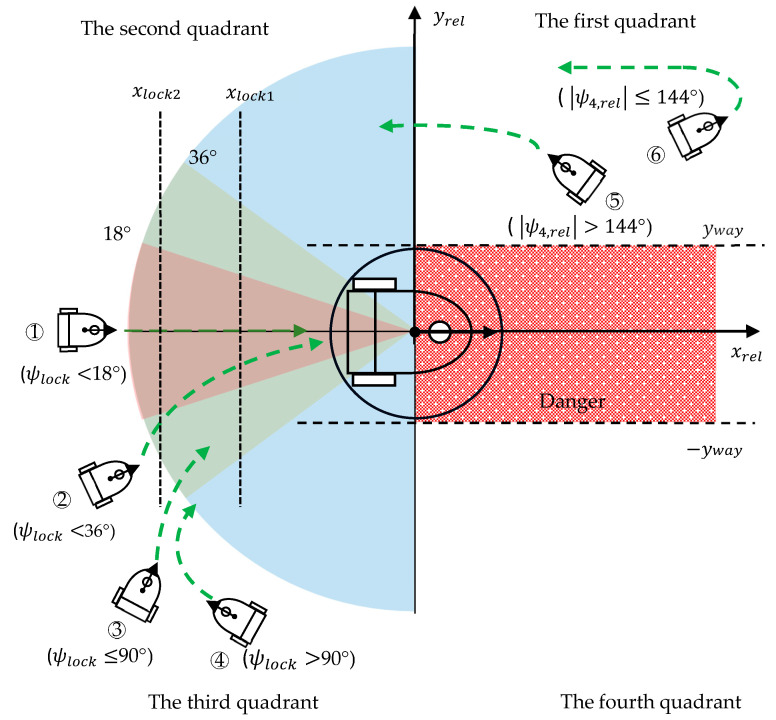
Survival penalty function that guides the WMR’s movement.

**Figure 7 sensors-23-08651-f007:**
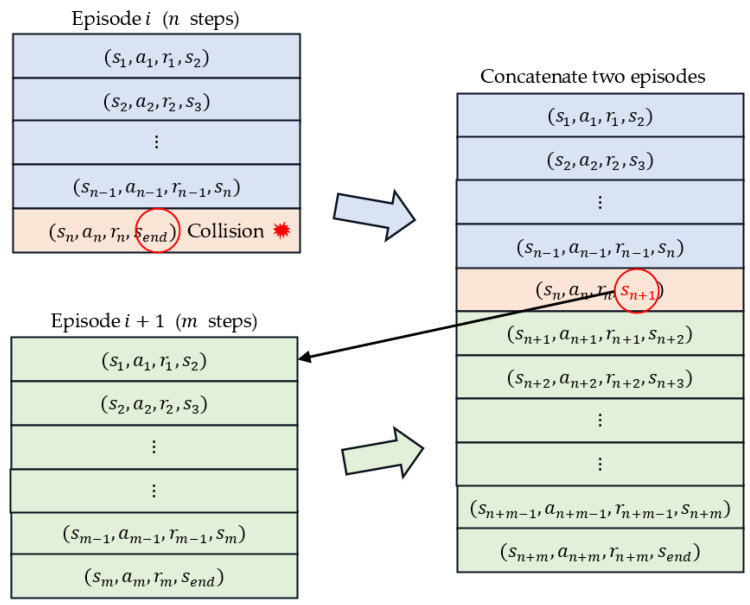
Preprocessing the replay buffer for obstacle avoidance. Red circle indicates the original next state $s_{end}$ is linked to the state of the first step of episode i + 1.

**Figure 8 sensors-23-08651-f008:**
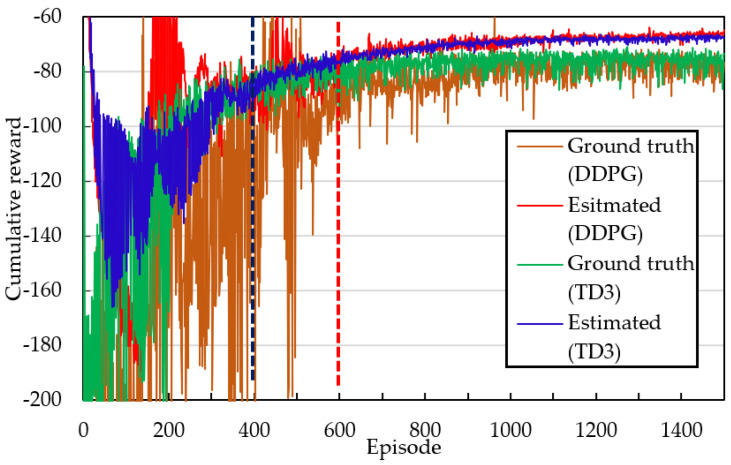
Comparison of cumulative rewards between DDPG and TD3 algorithms at yaw angle of 90°. Note: The vertical blue and red lines indicate that the estimated rewards calculated by the TD3 and DDPG algorithms gradually become stable and consistent after approximately 400 and 600 episodes of training, respectively.

**Figure 9 sensors-23-08651-f009:**
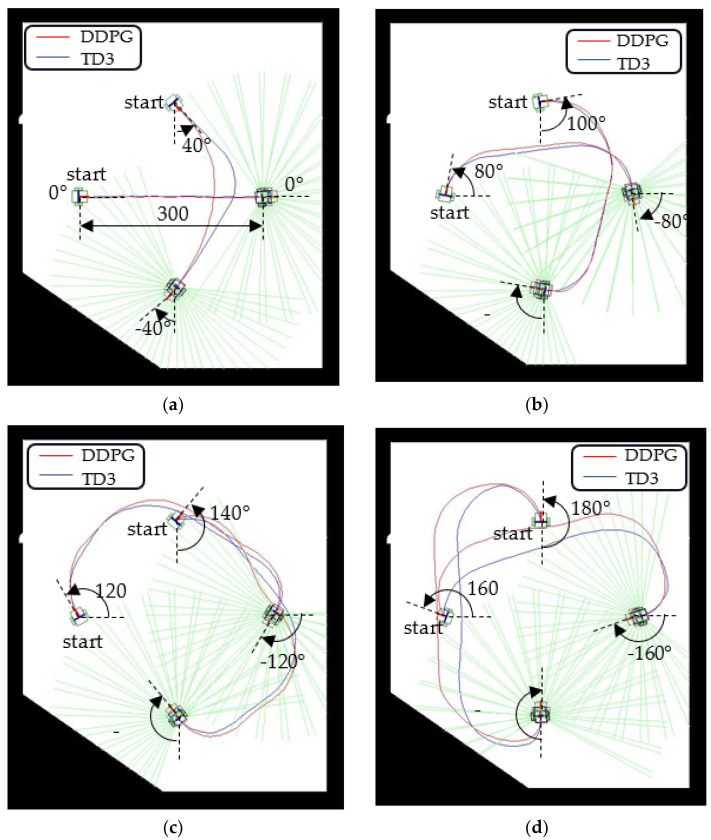
Navigation paths for various initial and target yaw angles: (**a**) yaw angles of 0° and 40°, (**b**) yaw angles of 80° and 100°, (**c**) yaw angles of 120° and 140°, and (**d**) yaw angles of 160° and 180°.

**Figure 10 sensors-23-08651-f010:**
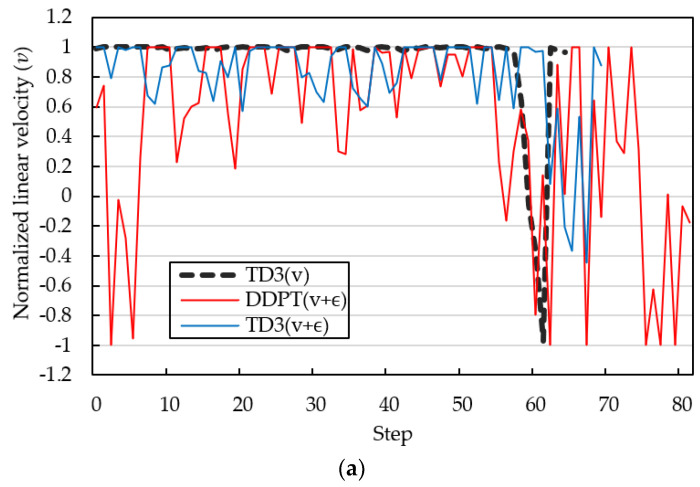
The trajectories of linear and angular velocities at yaw angle of 80°: (**a**) linear forward velocity and (**b**) angular velocity.

**Figure 11 sensors-23-08651-f011:**
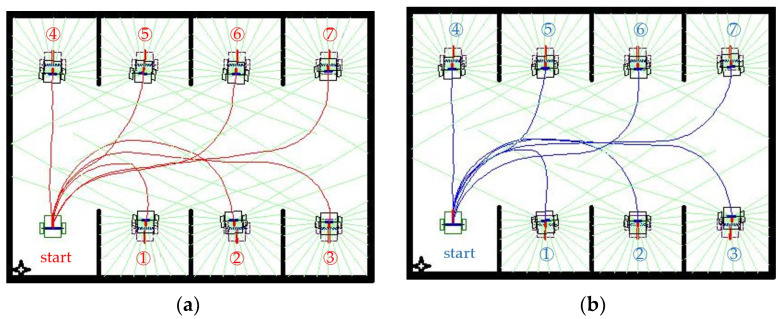
Path planning in a parking scenario when the DDPG and TD3 algorithms are used: (**a**) DDPG and (**b**) TD3. Note: Numbers indicate parking locations.

**Figure 12 sensors-23-08651-f012:**
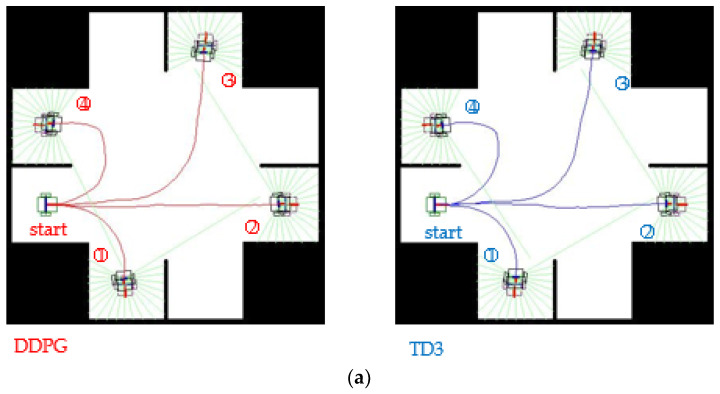
Routing plan for an intersection without and with an obstacle: (**a**) obstacle-free scenario and (**b**) scenario with an obstacle. Note: Numbers represent the exit numbers.

**Figure 13 sensors-23-08651-f013:**
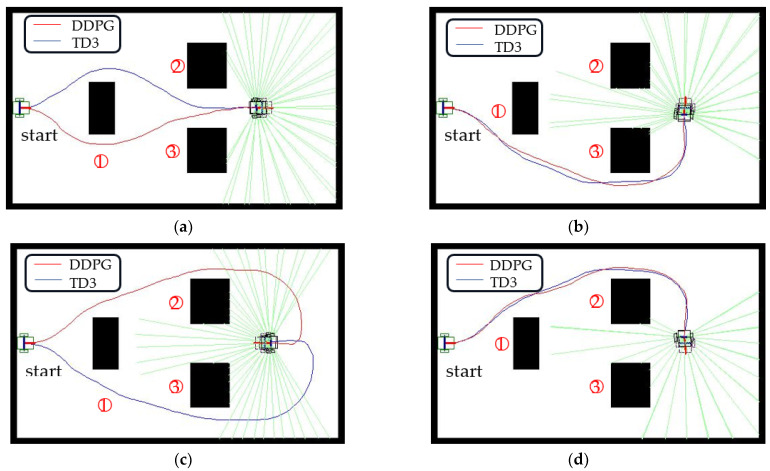
Routing plan for multiple obstacles: (**a**) $\psi_{goal}$ = 0°, (**b**) $\psi_{goal}$ = 90°, (**c**) $\psi_{goal}$ = 180°, and (**d**) $\psi_{goal}$ = 270°. Note: Numbers represent the obstacle numbers.

**Table 1 sensors-23-08651-t001:** Architectures of the actor and critic networks.

Layers	Actor Network		Critic Network	
	# Units	Activation	# Units	Activation
Input layer	27	Relu	27	Relu
Hidden layer 1	512	Relu	512 + 2	Relu
Hidden layer 2	512	Relu	512	Relu
Hidden layer 3	512	Hyper. tangent	512	Linear
Output layer	2		1	

**Table 2 sensors-23-08651-t002:** Hyperparameter settings in the simulation.

Hyperparameter	Symbol	Value
Discounter factor	γ	0.99
Learning rate of actor network	$\alpha_{\mu }$	0.00001
Learning rate of critic network	$\alpha_{Q}$	0.00001
Soft update rate	τ	0.01
Max. Epsilon-Greedy	$\epsilon_{max}$	1.0
Min. Epsilon-Greedy	$\epsilon_{min}$	0.5
Replay buffer size		40,000
Batch size		128
Max. episode		1500
Max. step per episode		300
Optimization		Adam

**Table 3 sensors-23-08651-t003:** Geometrical parameters of and constraints on the WMR.

Description	Symbol	Value
Car width	2 × b	24
Car length	l	30
Distance between center of mass and geometric center of platform	d	10
Driving wheel radius	r	5
Driving wheel width		4
Driving wheel distance		28
Max. alignment position error		3
Max. alignment orientation error		±5°

**Table 4 sensors-23-08651-t004:** Simulation results obtained for autonomous navigation under various yaw angle configurations.

Scenario	DDPG					TD3				
	Esti. Reward	G.-T. Reward	Δ Reward	Step	Succe. Rate	Esti. Reward	G.-T. Reward	Δ Reward	Step	Succe. Rate
0°	−8.41	−13.36	4.95	70.2	0.97	−10.86	−11.50	0.64	54.6	1.00
20°	−18.80	−23.14	4.34	64.3	0.71	−20.42	−21.87	1.44	55.9	0.96
40°	−30.93	−36.10	5.16	75.5	0.88	−33.01	−35.23	2.22	64.0	0.90
60°	−47.61	−56.10	8.49	77.4	0.93	−50.75	−56.15	5.40	69.3	0.95
80°	−54.75	−64.51	9.76	77.7	0.97	−57.50	−64.23	6.73	74.6	0.93
100°	−69.09	−77.86	8.77	87.2	0.90	−71.19	−77.06	5.87	86.2	0.97
120°	−82.86	−92.73	9.87	106.9	0.93	−85.47	−92.49	7.02	103.9	0.99
140°	−98.53	−112.12	13.59	122.3	0.96	−101.22	−110.84	9.62	118.3	0.99
160°	−113.84	−129.81	15.97	134.2	0.95	−115.89	−127.43	11.54	127.9	0.97
180°	−158.19	−179.89	21.71	151.8	0.65	−162.40	−180.08	17.68	146.7	0.89

**Table 5 sensors-23-08651-t005:** Simulation results obtained in the parking lot scenario.

Scenario	DDPG					TD3				
	Esti. Reward	G.-T. Reward	Δ Reward	Step	Succe. Rate	Esti. Reward	G.-T. Reward	Δ Reward	Step	Succe. Rate
1	−43.48	−54.96	11.48	60.09	0.93	−46.04	−54.90	8.86	58.76	0.98
2	−51.56	−61.94	10.38	63.47	0.90	−53.98	−61.55	7.57	59.40	0.96
3	−63.95	−75.31	11.36	82.63	0.86	−66.28	−74.99	8.70	77.61	0.96
4	0.45	−2.58	3.03	37.10	0.87	−1.93	−2.07	0.15	32.22	0.98
5	−20.90	−25.29	4.39	43.04	0.94	−23.25	−25.25	2.00	41.44	0.97
6	−41.30	−47.83	6.53	66.36	0.92	−43.63	−47.82	4.19	61.79	0.97
7	−56.99	−65.60	8.61	87.21	0.90	−59.88	−65.58	5.70	78.10	0.93

**Table 6 sensors-23-08651-t006:** Simulation results obtained in the intersection scenario.

Scenario	DDPG					TD3				
	Esti. Reward	G.T. Reward	Δ Reward	Step	Succe. Rate	Esti. Reward	G.T. Reward	Δ Reward	Step	Succe. Rate
1	−13.33	−17.12	3.79	32.04	0.89	−14.86	−18.01	3.15	32.81	0.97
2	−8.76	−12.02	3.26	56.99	0.77	−10.35	−11.60	1.26	54.63	0.99
3	−30.49	−37.51	7.02	65.37	0.84	−32.54	−37.57	5.03	63.45	0.96
4	−43.74	−54.09	10.35	56.99	0.89	−45.76	−54.48	8.72	57.39	0.97
1 *	38.53	−18.01	56.54	38.31	0.84	−15.57	−17.49	1.92	33.67	0.96
2 *	33.60	−14.93	48.53	61.27	0.74	−13.87	−14.17	0.29	55.72	0.97
3 *	−4.05	−53.84	49.79	67.86	0.64	−50.54	−54.51	3.97	65.68	0.80
4 *	1.76	−55.38	57.14	62.21	0.86	−46.76	−54.81	8.05	57.12	0.99

Asterisks represent scenarios containing a circular obstacle.

**Table 7 sensors-23-08651-t007:** Simulation results obtained in the multi-obstacle scenario.

Scenario	DDPG					TD3				
	Esti. Reward	G.T. Reward	Δ Reward	Step	Succe. Rate	Esti. Reward	G.T. Reward	Δ	Step	Succe. Rate
0°	−27.57	−35.01	7.44	92.68	0.96	−31.30	−33.18	1.88	77.16	1.00
90°	−84.27	−97.32	13.04	95.92	0.98	−88.93	−95.07	6.14	94.23	0.98
180°	−119.18	−135.27	16.10	148.69	0.94	−126.30	−134.62	8.32	127.15	0.96
270°	−82.85	−99.34	16.50	147.56	0.81	−90.14	−95.45	5.31	94.09	0.92

## Data Availability

Data available upon request.

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
