# Peer review of "End-to-End Autonomous Navigation Based on Deep Reinforcement Learning with a Survival Penalty Function"

_sensors, 2023, doi:10.3390/s23208651_

Round 1

Reviewer 1 Report

The method proposed in this manuscript has some innovative content, but the analysis of previous studies is not accurate enough.

1. It is recommended that the author supplement the summary of previous research.

2. The environment in the manuscript is relatively simple. Can you add some more realistic environments. Because the performance of navigation methods is closely related to environments.

3. The author should conduct comparative testing with some existing classic methods.

4. Please discuss the rationality of the survival penalty function? Are there any other design methods?   5. What is the potential application value of this study compared to other previous studies?   6.What adjustments are needed when applying this method to robots with other motion structures?

There are some grammar errors in the manuscript, it is recommended to polish it.

Reviewer 2 Report

1.     Line 173, “left and right” should “right and left”.

2.     Eq. (25), “-lock2” should be corrected.

3.     The title of Fig. 9(a) should be corrected.

4. The authors should survey more papers related to reinforcement learning-based obstacle avoidance for mobile robots and then discuss the contribution of this paper.

5. Please provide the references for Sec. 3.1 DDPG Algorithm and Sec. 3.2 TD3 Algorithm, respectively.

6. Please provide a figure of trajectories of the action v and ω in free obstacle simulation.

7. To better verify the proposed method, it is suggested that the virtual environment should contain multiple obstacles for navigation simulation.

Reviewer 3 Report

This is a good English paper. This study proposes an end-to-end autonomous navigation method based on deep reinforcement learning, using two algorithms: DDPG and TD3. It introduces a comprehensive reward mechanism based on a penalty function, which effectively solves the sparse reward problem. It also performs simulation verification in three scenarios. Reading the following suggestions carefully will help you further improve your manuscript.

1.      Introduction

---The introduction section provides a general overview of two categories of methods: traditional and learning-based, and summarizes the relevant research on path planning in a comprehensive manner. However, there are some aspects that could be enhanced:

a)        When discussing the traditional methods for planning algorithms, it would be beneficial to include more examples of such methods, such as graph search algorithms (Dijkstra, A*), sampling-based algorithms (PRM, RRT), interpolation curve algorithms, reactive algorithms (DWA), and so on. This would make the survey of traditional methods more complete and informative. Source from literature:

Zhou, C.; Huang, B.; Fränti, P. A review of motion planning algorithms for intelligent robots. Journal of Intelligent Manufacturing. 2022, 33(2), pp. 387-424.

b)        I think the literature review section could be more comprehensive. Besides the classification of methods, it should also discuss the current research from the perspectives of application scenarios, problem difficulties, and other aspects that are more relevant to the main topic of this paper.

c)        I suggest that the introduction section of this paper should only briefly introduce the work content, and the detailed innovation and contribution should be placed in the discussion section. This would make the structure and logic of the paper more clear and coherent.

2.      Problem Statement and Preliminaries

---The robot path planning research in this paper is based on a two-dimensional framework of x and y directions, but in reality, the height change in the z direction is also a significant influencing factor. What do you think about extending the existing two-dimensional path planning algorithm to three dimensions? Assuming that extending to three dimensions has great prospects for autonomous navigation of flying robots such as drones, I hope you can tell me your thoughts.

3.      Simulation and Results

---The two vertical dashed lines in Figure 8 indicate the start and end points of the path planning algorithm. The black line represents the initial position of the robot, and the red line represents the goal position of the robot. The figure caption should include this information to make the figure more clear and understandable.

---Line 467, this paragraph only gives the result that the TD3 algorithm is overall better than the DDPG algorithm, but does not analyze the possible reasons. Please provide corresponding analysis based on the principles of the two algorithms, such as from the aspects of algorithm structure design complexity, training strategy, etc. In addition, the same problem applies to the other two scenarios.

---I hope the author can add a discussion section, not only to compare the differences between the two algorithms in three scenarios, but also to compare and analyze the impact of the scenarios on the algorithm adaptability, and to discuss in depth from multiple perspectives. The following are some questions for discussion:

a)        The strengths and weaknesses of the two algorithms in different scenarios, and the possible reasons for their performance differences.

b)        The challenges and difficulties of each scenario, and how they affect the algorithm adaptability and robustness.

I congratulate the author on achieving innovative results.

Round 2

Reviewer 1 Report

Thank you for the detailed modifications made by the authors. I think all my questions have been answered in this new version.

The English of this paper is fluent and logically clear.

Reviewer 3 Report

Dear, authors

Thank for providing revised version and taking into account my comments. I can see a lot of improvements compared to the first version of the manuscript. I think it can be published in present form.